A time-lapse photography method for monitoring salmon (Oncorhynchus spp.) passage and abundance in streams

Deacy William W. william.deacy@umontana.edu will.deacy@gmail.com 1
Leacock William B. 2
Eby Lisa A. 3
Stanford Jack A. 1
1 Flathead Lake Biological Station, University of Montana , Polson , MT , USA
2 Kodiak National Wildlife Refuge, US Fish and Wildlife Service , Kodiak , AK , USA
3 Wildlife Biology Program, University of Montana , Missoula , MT , USA
Johnson Magnus
Electronic publication date: 2016 Jun 14
Publication date: 2016
Volume: 4
Electronic Location ID: e2120
Received 2016 Feb 28; Accepted 2016 May 19
Copyright: ©2016 Deacy et al.
Copyright year: 2016
Copyright holder: Deacy et al.
License: This is an open access article distributed under the terms of the Creative Commons Attribution License, which permits unrestricted use, distribution, reproduction and adaptation in any medium and for any purpose provided that it is properly attributed. For attribution, the original author(s), title, publication source (PeerJ) and either DOI or URL of the article must be cited.
License URL: https://creativecommons.org/licenses/by/4.0/

Keywords: Salmon, Time-lapse photography, Weir, Spawning, Kodiak, Alaska, Sockeye, Migration, Video, Escapement

Funding: Jessie M. Bierman Professorship at Flathead Lake Biological Station The US Fish and Wildlife Service Youth Initiative, Refuge, and Inventory and Monitoring Programs Funding for this work was provided by the Jessie M. Bierman Professorship at Flathead Lake Biological Station and the US Fish and Wildlife Service Youth Initiative, Refuge, and Inventory and Monitoring Programs. The funders had no role in study design, data collection and analysis, decision to publish, or preparation of the manuscript.

==============================
Accurately estimating population sizes is often a critical component of fisheries research and management. Although there is a growing appreciation of the importance of small-scale salmon population dynamics to the stability of salmon stock-complexes, our understanding of these populations is constrained by a lack of efficient and cost-effective monitoring tools for streams. Weirs are expensive, labor intensive, and can disrupt natural fish movements. While conventional video systems avoid some of these shortcomings, they are expensive and require excessive amounts of labor to review footage for data collection. Here, we present a novel method for quantifying salmon in small streams (<15 m wide, <1 m deep) that uses both time-lapse photography and video in a model-based double sampling scheme. This method produces an escapement estimate nearly as accurate as a video-only approach, but with substantially less labor, money, and effort. It requires servicing only every 14 days, detects salmon 24 h/day, is inexpensive, and produces escapement estimates with confidence intervals. In addition to escapement estimation, we present a method for estimating in-stream salmon abundance across time, data needed by researchers interested in predator--prey interactions or nutrient subsidies. We combined daily salmon passage estimates with stream specific estimates of daily mortality developed using previously published data. To demonstrate proof of concept for these methods, we present results from two streams in southwest Kodiak Island, Alaska in which high densities of sockeye salmon spawn.

Introduction

The management, research, and conservation of salmonids requires accurate estimations of population sizes. Managers use salmon (Oncorhynchus spp.) escapement estimates (salmon remaining after harvest that enter freshwater to spawn) to develop stock-recruit curves and to improve management timing of fisheries. Researchers often need escapement data for studies involving productivity, nutrient subsidies, and predator--prey dynamics. Although we have good escapement data for many main-stem rivers used by migrating salmon, we have little escapement data at smaller scales, including small streams where many salmon ultimately spawn (Anadromous Waters Catalog, Alaska Department of Fish and Game). This is regrettable given that large salmon stock-complexes are composed of dozens or hundreds of distinct salmon populations, many of which spawn in first and second order streams (Hilborn et al., 2003). Collections of small salmon populations spawning at different times and in different locations tend to have more stable interannual abundance than a single homogenous population due to ``portfolio effects,'' which results in more reliable returns and fewer closures for commercial fisheries (Schindler et al., 2010). This stability arises from population diversity occurring at small spatial scales (i.e., first and second order streams), hence the importance of having the tools to investigate and understand these populations to effectively manage salmon for human and wildlife consumers.

Watershed-scale escapement estimates do not effectively characterize the resources available to wildlife consumers, because they do not tell us how long salmon are available to consumers. In many watersheds, consumers cannot catch salmon while they migrate up the relatively deep water of main-stem rivers; they must wait until salmon enter shallow spawning streams where they are more easily caught. As a result, consumers interact with individual salmon populations rather than entire stock complexes, and thus, watershed scale escapement can be a poor estimate of the salmon available to consumers of conservation concern such as eagles (Levi et al., 2015), bears (Deacy et al., 2016), and trout (Bentley, Schindler & Armstrong, 2012). Consumers are easily satiated by even modest densities of spawning salmon, so the duration of spawning activity is likely just as important to consumers as the abundance of salmon (Jeschke, 2007). Despite the importance of small tributary salmon escapement to salmon management, ecosystem function, and salmon conservation, existing methods of monitoring salmon abundance do not perform well at these sites, given high financial costs, investment of time, and the potential to alter salmon behavior.

Traditionally, anadromous salmonids (Oncorhynchus spp.) moving into large rivers or streams have been counted by observers stationed at fish weirs, fences, and observation towers, or by use of sonar stations (Table 1; Cousens et al., 1982; Johnson et al., 2007). These methods can produce reliable estimates; however, high labor and equipment costs make them too expensive for simultaneously monitoring many streams. To fill this gap, researchers have experimented with systems that record video of passing salmon using either under or above water cameras (Hatch, Schwartzberg & Mundy, 1994; Davies, Kehler & Meade, 2007; Van Alen, 2008). These video weir methods have three key advantages: (1) footage can be reviewed long after the data are collected, allowing a small crew to monitor several runs simultaneously; (2) periods with high salmon abundance can be counted more accurately by reducing playback speed; and (3) fewer site visits reduce impacts on wildlife caused by human presence. Although these benefits have made video enumeration an increasingly popular method for counting salmonids, reviewing large amounts of video is required. The resulting personnel costs make video weir methods impractical for many applications. A method is needed for collecting escapement data that produces reliable estimates without thousands of hours of video review or frequent site visits. Furthermore, some enumeration methods (i.e., weirs) can obstruct natural movements of salmon and other fishes. This may not be a problem on main-stem rivers if salmon tend to move consistently upstream, however, it is problematic in small streams where diel movements into and out of streams are common (Bentley et al., 2014).

Table 1 Comparison of salmon enumeration methods.

	Method	Typical sites	Advantages	Disadvantages	References	
Real time counts	Weir	Large clear rivers/streams	Easy sampling of age, sex, length, genetics	Expensive (equipment/personnel); May hinder natural fish movements	Anderson & McDonald (1978)	
Observation tower	First to fifth order clear streams/rivers	Does not hinder fish passage	Expensive (personnel); turbulence or bad light can make counts difficult	Cousens et al.(1982)	
Retrospective counts	Sonar	Large clear or opaque rivers	Not affected by turbulence; Records of run can be saved and reviewed; Playback can be slowed and counts repeated for QA/QC; Does not obstruct fish passage	Expensive (equipment/personnel); Lengthy footage review; Accuracy suffers at highest densities	(Holmes et al., 2006; Maxwell & Gove, 2007)	
Video net weir	Medium to small rivers and streams	Records of run can be saved and reviewed; Playback can be slowed and counts repeated for QA/QC; Does not obstruct fish passage	Expensive (equipment/personnel); Lengthy footage review; May hinder natural movements of fish; Nets can catch debris	Van Alen (2008), Grifantini et al. (2011)	
Above water video	Medium to small clear streams	Records of run can be saved and reviewed; Playback can be slowed and counts repeated for QA/QC; Does not obstruct fish passage	Expensive; Time consuming footage review	Hatch, Schwartzberg & Mundy(1994)	
Time-lapse double sampling	Medium to small clear streams	Inexpensive; Can be left unattended for 14 days; Records of run can be saved and reviewed; Playback can be slowed and counts repeated for QA/QC; Does not obstruct fish passage; decreased impacts on wildlife	Limited to smaller streams (<15 m)	Current study	

In addition to total escapement, studies focused on consumer responses to availability of salmon need to know the number of living salmon in streams (hereafter in stream abundance) across time. In stream abundance across time represents foraging opportunities better than gross escapement when consumers are swamped by a pulsed resource, which is often the case for consumers of spawning salmon (Armstrong & Schindler, 2011). Typically, in stream abundance data are collected using ground (Quinn et al., 2001) or aerial surveys (Neilson & Geen, 1981) which are repeated several times during a salmon run. Ground surveys work well on streams that are easy to access, small enough to survey in a reasonable amount of time, and where disturbing wildlife is not a concern. Aerial surveys may work well for less accessible sites if visibility from the plane is not impeded by riparian vegetation or complex channel geomorphology. Moreover, because salmon abundance in streams tends to change rapidly, these methods only work well when the survey frequency is high. Furthermore, to collect reliable data using aerial surveys, researchers need to correct for differences among observers (Bue et al., 1998). Here, we present an alternative method for estimating the number of living salmon in a stream through the full duration of the run. The approach combines daily estimates of salmon passage, collected using our time-lapse camera system, with a model of spawning salmon mortality.

Our system requires service only every 14 days, detects salmon 24 h/day, is inexpensive to implement, and produces escapement estimates with confidence intervals. This system works on rivers and streams up to ∼15 m wide and ∼1 m deep. In addition, we present a method for estimating in stream salmon abundance, data which are important for studies focused on the response of wildlife consumers to salmon runs and nutrient subsidies. To demonstrate proof of concept, we present results from two small streams with very high densities of salmon.

Methods

Approach

To harness the advantages of remote camera systems without time-consuming video enumeration, we utilized a ``double sampling'' scheme, which is often used when a variable of interest is costly to measure, but an auxiliary variable is more easily measured and has a predictable relationship to the variable of interest (Cochran, 1977). The cheaper variable can be measured for all of the sample units while the expensive variable is measured on a subsample of units in order to model the relationship between the variables. Here, our variable of interest is the number of salmon that pass into and out of a stream each hour, which we can accurately quantify with an above-water video camera. The related auxiliary variable is the number of salmon detected in time-lapse images each hour. The total time required to review footage is low relative to video-only approaches because we only have to enumerate salmon in a subset of the hour long sample units. We determined the salmon passage for the remaining hours by modelling the relationship between the subsample of hourly video counts and photo counts and then using the model to predict salmon passage across the entire salmon run.

Study streams

We developed this method on two streams used by spawning sockeye salmon: Meadow and Southeast Creeks (Fig. 1A) in southwest Kodiak Island, Alaska. Meadow Creek is a second order tributary to Karluk Lake. It has a mean width of 4.50 m and depth of 13 cm in the lower 0.8 km used by spawning salmon. Southeast Creek is a first order tributary to Red Lake that flows out of a small spring pond. It has a mean width of 3.90 m and depth of 9.1 cm in the lower 2.7 km used by spawning salmon. Salmon enter these streams annually to spawn and bidirectional movement and pre-spawn mortality is common owing to a large number of brown bears that prey on the salmon.

Figure 1 Map and pictures of counting system.

(A) Map of southwest Kodiak Island, Alaska showing the locations of streams where salmon were counted using time-lapse double sampling. (B) Salmon counting system including foldable steel tower holding a time-lapse camera (box at top), video camera, and solar panels. The tower is surrounded by an electric fence to prevent equipment disruption by bears. White high density polyethylene (HDPE) plastic panels were placed on the stream bed to improve sightability of sockeye salmon (Oncorhynchus nerka). (C) An image of two sockeye salmon passing across the contrast panels, with the video camera in the foreground.

Time lapse camera system

To record time lapse images of passing salmon, we used a Reconyx® Hyperfire PC800 camera, programmed to take three photos in rapid succession (<1 s between frames) each minute, 24 h/day. Each three frame burst allowed us to detect the number and direction of travel (up or downstream) of salmon passing the camera. We suspended the time lapse camera above the stream using steel electrical conduit attached to a steel Big Game® Pursuit tripod tree stand positioned adjacent to the stream, approximately 50 m upstream of the lake (Fig. 1B). We attached the camera to the conduit with a Camlockbox® ball mount which allowed us to easily aim the camera. To light the streambed at night we secured an LTS® IR50 850 nm infrared (IR) light to the tripod platform. Although visible light would have worked well, we used IR light to avoid changing the behavior of salmon and/or their predators with visible light. The Reconyx camera and infrared light were powered by an 80 amp-hour deep-cycle battery charged by a 100 W solar panel secured to the south side of the tower.

To record video, we secured a video camera to the top of the tower. The video footage was stored by a Digital Video Recorder (DVR) set to record D1 resolution, 30 frames per second video from 12 pm to 8 pm, the periods with the best quality video (good light) and the majority of salmon movement activity. The video camera and DVR were powered by its own battery/solar power system, identical to the one powering the Reconyx camera and IR light. To make passing salmon easier to see, we secured 50.8 cm × 76.2 cm white High Density Polyethylene (HDPE) contrast panels to the bottom of the stream below the cameras by attaching them to a heavy chain (Alaska Department of Fish and Game Permit # FH-14-II-0076). The HDPE panels are buoyant in water and the chain prevents the panels from floating off of the streambed. Using stainless steel carabineers, we attached the chain to T-posts which we pounded into the margins of the streambed. To prevent salmon from swimming under the panels, we pinned the chain to the stream bed using several steel stakes.

We visited each camera system every two weeks from early June through early September to switch out data cards and remove algae and debris from the contrast panels. Back at our field station, we separately counted the number of salmon moving up and downstream past the contrast panels during each three-photo burst. We only counted a salmon as passing if it moved at least 1/2 the length of the panels; we did not count stationary fish. Finally, we summed upstream and downstream counts separately for each hour of the monitoring season. To ensure consistent counting technique, each stream was counted by the same person for the entire season.

Modelling salmon escapement (abundance)

We used a model-based double sampling approach to estimate salmon escapement. We modelled the relationship between video salmon counts and photo salmon counts for a non-random subsample of hours, and then used this model to predict salmon passage for the entire season. This is different from the ``sampling-design approach'' more commonly used to double sample (Cochran, 1977). If we had used the sampling-design approach, we would have counted the salmon passing in a simple random subsample of video hours, and then calculated the total escapement by multiplying the time lapse salmon count by the ratio of video counts to photo counts in the subsample. However, the sampling design-based approach has two requirements which are difficult to satisfy. First, to be random, every hour of the salmon run must be available for sampling, meaning that video must be recorded throughout the entire run. A single day of missed video (due to a power outage, insects sitting on the lens, etc.) could significantly bias the resulting abundance estimates if the outage occurred on a day with relatively few or many passing salmon. Second, the video must be high enough quality to assume 100% salmon detection. This requirement can be difficult to meet because of glare and poor night-time video quality. Rather than attempt to design a system that meets these strict requirements, we used a model-based approach, where we model the relationship between video counts and time lapse counts (Stephens et al., 2012). This framework allows us to select our sample of video-enumerated hours non-randomly; our estimate of abundance is unbiased as long as the model is correctly specified (Hansen, Madow & Tepping, 1983; Gregoire, 1998).

We selected 70 h that spanned the full range of hourly time-lapse salmon counts, from the hours with many salmon swimming downstream to hours with strong upstream movement. Also, we selected hours where we were confident of nearly 100% detection, excluding hours with bad glare or poor lighting. In total, we watched 70 h of video for each stream, however, because we considered up and downstream salmon movement independently, this gave us a sub-sample of 140 values for each stream (70 upstream counts and 70 downstream counts).

Next, we modelled video counts as a function of time-lapse photo counts for the subsample. All statistical analyses were conducted using the statistical program R 3.1.3 (R Development Core Team, 2015). We compared four different models for each stream: first and second order linear regressions and first and second order segmented or ``split-point'' linear regressions (Muggeo, 2008) (Table 2). The segmented regression allows the slope to differ across ranges of the predictor variable. This makes sense for salmon swimming in a stream; salmon swimming upstream (positive values) might move slower, and thus have a greater chance of being detected in a time-lapse burst. In contrast, salmon swimming downstream (negative values) might move faster and have a lower likelihood of detection. To address this possibility, we including segmented regression models with the split-point (slope inflection point) constrained to zero. To assess relative model fit, we compared Akaike's Information Criterion values (AICc; Akaike, 1974). To validate models and test for over-fitting, we performed leave one out cross validation (LOOCV; Kohavi, 1995), and used the resulting predictions to calculate the precision (mean squared error, MSE) and accuracy (the percent difference between the predicted and actual escapement of the 70 h for which we watched video). Based on these metrics, we selected a top model for each stream.

Table 2 Details of salmon passage models.

Model descriptions, escapement estimates and model validation metrics. AICc is Aikaike's Information Criterion adjusted for small sample size (Akaike, 1974). The 95% confidence intervals on escapement were calculated using bootstrap resampling methods. Accuracy is the percent difference between the leave-one-out cross-validation predicted escapement and the actual escapement for the 70 h where escapement was counted using video recording. Precision is the mean squared error (MSE). The top model for each stream is in italics.

	Model name	Model of sockeye passage	k	AICc	Escapement estimate (±95% CI)	Accuracy	Precision	
Meadow	Segmented first order	pass =(x > 0)∗15.242 + (x < 0)∗18.920	2	1534.30	30,509 (±9,494)	+3.0%	3,556	
Segmented polynomial	pass =(x > 0)∗16.096 + (x > 0)2∗ − 0.0206 + (x < 0)∗18.454 + (x < 0)2∗ − 0.0206	3	1535.02	30,064 (±14,211)	+10.1%	3,557	
First order	pass =x∗16.1441	1	1551.56	41,539 (±3,692)	+37.5%	3,917	
Polynomial	pass =x∗17.3137 + (x2∗ − 0.04552)	2	1535.94	31,830 (±11,628)	+24.4%	3,591	
Southeast	Segmented first order	pass =(x > 0)∗22.78 + (x < 0)∗22.18)	2	1733.43	68,253 (±15,759)	+7.2%	14,932	
Segmented polynomial	pass =(x > 0)∗22.555 + (x > 0)2∗0.0045 + (x < 0)∗22.653 + (x < 0)2∗ − 0.0045	3	1735.26	66,303 (±23,052)	+5.7%	15,569	
First order	pass =x∗22.505	1	1732.17	65,355 (±4,305)	+3.1%	14,167	
Polynomial	pass =x∗22.589 + x2∗ − 0.0040	2	1733.44	66,610 (±8,045)	+6.2%	14,669	

Using the best model for each stream, we predicted the salmon passage for all of the hours of the monitoring period. The sum of these predictions is the estimated escapement. Because we did not use random sampling to select our modelling subsample, it is inappropriate to use the model variance to calculate confidence intervals for total escapement. Instead, we bootstrapped our subsample (Canty & Ripley, 2016) with replacement (140 values to match our original subsample), refit our model using the top model structure, and re-predicted the total escapement (Efron & Tibshirani, 1993). We repeated this 10,000 times and used the 2.5 and 97.5 percentile values as upper and lower 95% confidence intervals of total escapement.

Modelling number of living salmon in streams

To model in stream abundance across time, we took daily escapement estimates (upstream moving salmon minus downstream moving salmon), and applied mortality estimates from the literature. Carlson et al. (2007) investigated the relationship between stream width/depth and stream life (number of days from salmon stream entry to death) on a range of tributaries to Nerka and Aleknagik Lakes, Alaska which are morphologically similar to our focal streams. The three main sources of mortality for spawning sockeye salmon were senescent death, predation (mostly by bears), and stranding. They found that salmon spawning in wider/deeper streams tended to have longer stream lives. The authors' explanation was that salmon in shallow/narrow streams experienced higher predation rates which selects for more rapid reproductive cycles and consequently earlier deaths. Because of this interaction between stream morphology and salmon stream life, it is probably inappropriate to use a single estimate of stream life across streams with varying morphology. We used the results of Carlson et al. (2007; Table 1) to create a model of stream life as a function of stream morphology.

Assuming salmon in our streams were equally likely to die by stranding, predation, and senescence as they were in the Carlson study, we calculated a weighted average of the mean stream life (weighted by the number of salmon dying by each mechanism) for each of the Carlson et al. (2007; Table 1) streams. We then used this weighted average stream life as the response variable and stream width and depth as predictor variables in a simple linear regression model. Because stream depth and width were strongly correlated (r = 0.90), including both variables in the model resulted in collinearity. We thus selected between depth-only and width-only models by comparing AICc scores. We then used the top model to predict the mean stream life of sockeye salmon in Meadow and Southeast Creek, using field measurements of stream morphology measured in 2014 as predictors. There was a strong positive correlation between the mean and pooled standard deviation (Hedges, 1981) of stream life in the Carlson data (r = 0.95, p = 0.004); therefore, rather than model the standard deviation (SD) of stream life separately from the mean, we assumed stream life SD was proportional to the mean (SD = 0.499 * mean stream life).

To calculate in stream abundance each day, we summed the number of salmon that entered on that day with the predicted number of surviving salmon from the previous days: Living Salmon on Dayx= ∑t=1NPx+Px−tSt

where Px is the number of salmon that passed into the stream on day x, Px−t is the number of salmon that passed into the stream t days before day x, St is the proportion of those salmon surviving to day x, and t is an index of days. The values of St are from the cumulative distribution function of survival which we modelled above. N is the number of days it takes for survival (St) to reach zero, which varies based on the survival model (it will be larger on deeper streams where stream life is greater).

To understand the sensitivity of in stream abundance models to changes in stream life estimates, we calculated in stream abundance for each stream across a range of stream life values. We then used percent change in maximum abundance to assess the impact of changing stream life. Because the amount of time consumers have access to salmon is at least as important as peak abundance, we also calculated the duration of the salmon run, defined as the number of days where abundance was at least ten percent of the maximum in stream estimate from the un-altered model. This (admittedly arbitrary) ten percent threshold was an attempt to set a lower limit on the salmon density below which benefits to consumers decline.

Results

Salmon escapement

Of the suite of models relating video counts to time lapse counts for Meadow Creek, the top model was the segmented first order model (Table 2 and Fig. 2). It had the lowest AICc (1534.3), best precision (MSE = 3,556), and best accuracy (+3.0%). The segmented models likely explained more variation than the unsegmented models because salmon had different detection rates while swimming upstream versus downstream (salmon swim slower against the current), in the relatively steep gradient of Meadow Creek. Using the top model, the predicted escapement for Meadow Creek was 30,509 ± 9,494 (95% confidence intervals).

Figure 2 Relationship between hourly time-lapse and video counts of salmon passage for two streams in southwest Kodiak Island, Alaska.

The lines show the top model for each stream, selected using Akaike's Information Criterion adjusted for small sample size (AICc) (Akaike, 1974): a segmented first order relationship for Meadow Creek, and a simple first order linear relationship for Southeast Creek. The segmented model (Meadow) has a different slope above and below the origin, which is indicated by crossed vertical and horizontal dashed lines.

The top model for Southeast Creek was the first order regression which had the lowest AICc (1732.2), best precision (MSE = 14,167), and best accuracy (+3.1%). In contrast to Meadow Creek, the segmented model only explained slightly more variation than the first order model, but required an additional parameter. This suggests salmon in Southeast Creek have a similar detection rate whether they are swimming up or downstream, which is likely because Southeast Creek has a relatively flat gradient and low velocity. The total escapement for Southeast Creek was 65,355 ± 4,305 (95% confidence intervals). For Southeast Creek, the escapement estimates were not very sensitive to the model selected (maximum difference of only 4.4%) (Fig. 3). This contrasts with Meadow Creek, where the difference between the highest and lowest estimate was 38%.

Figure 3 In-stream salmon abundance calculated using four different models.

Comparison of estimates of the number of living sockeye salmon in two streams derived using four different models (model details in Table 1).

Modelling number of living salmon in streams

The model with depth as a predictor (AICc=27.5) explained more variation than the width model (AICc=31.9), so we used this model to predict mean stream life for our two streams. Meadow Creek had a predicted mean stream life of 7.1 days (SE = 3.5) while Southeast Creek (which is shallower), had a predicted stream life of 5.9 days (SE = 3.0). Using these values, we found the predicted salmon abundance over time in each stream were quite different; abundance peaked at just over 11,000 sockeye on July 11th in Meadow Creek and the run was finished around August 16th (Fig. 4). In contrast, Southeast Creek had two distinct peaks in abundance: the first on July 21st with just over 15,000 sockeye and the second peaking at 4,645 on August 29th. Thus, although the total escapement in Southeast Creek was more than double that of Meadow Creek, the peak salmon abundance was only 29% higher in Southeast.

Figure 4 Estimated salmon passage, escapement, and in-stream abundance.

Estimated hourly sockeye salmon passage (A, B), estimated cumulative passage (C, D), and estimated in-stream salmon abundance (E, F) in Meadow and Southeast Creeks. In the salmon passage plots (E, F), positive numbers indicate salmon moving into the stream from the downstream lake, while negative numbers indicate salmon leaving the stream and entering the lake.

In general, the in stream abundance models were quite sensitive to changes in stream life estimates. Increasing mean stream life in Meadow Creek by two days, from 7.1 to 9.1 days, increased the estimated maximum abundance by 14% (Fig. 5). The effect was even greater on Southeast Creek, with a 22% increase in abundance from a two-day increase in mean stream life. Increasing the standard deviation had the opposite effect: a one-day increase in SD of stream life decreased the maximum abundance by 5% and 3% on Meadow and Southeast Creeks, respectively. The sensitivity of salmon run duration (defined as the number of days with at least 10% of the maximum salmon abundance), to changes in mean and SD of stream life was less clear. On Meadow Creek, increasing mean stream life by two days increased the salmon run duration by two days (from 40 to 42 days) and increasing stream life SD by one day resulted in no measurable increase in salmon run duration. In contrast, the same changes on Southeast Creek resulted in a 5 day and 2 day increase in salmon run duration for changes to the mean and SD of stream life, respectively. This difference is likely because Southeast Creek has two distinct peaks in salmon abundance, and a two-day increase in stream life is a larger proportional change compared to Meadow Creek.

Figure 5 Sensitivity of in-stream salmon abundance estimates to changes in stream life.

Effect of altering mean and standard deviation (SD) of sockeye salmon survival (in days) on the estimated in-stream salmon abundance in Meadow and Southeast Creeks. In (A) and (B) the mean was manipulated, while the SD was altered in (C) and (D). In all plots, the unaltered model is shown in black.

Discussion

Researchers and managers increasingly acknowledge the important role of small salmon populations in generating stable returns for commercial fisheries and for supporting wildlife of high economic and commercial value (Schindler et al., 2010; Beacham et al., 2014). Many existing salmon monitoring tools were designed primarily for large streams and rivers and are ineffective or too expensive for monitoring the salmon populations that use small streams for spawning. The time-lapse salmon counting system presented here proved to be a low-cost, time-efficient, and accurate method for counting salmon in streams less than 15 m wide. This method only required bi-weekly site visits, which is ideal for remotely monitored sites and studies involving the response of wildlife to spawning salmon. These benefits will allow managers and researchers to quantify salmon in streams where it was previously too difficult or expensive. In addition, we presented a method for estimating the number of living salmon in a stream across the run, data which are particularly important for consumer-resource studies.

To estimate in stream salmon abundance, we developed a model of salmon stream life (number of days a salmon survives following spawning stream entry) based upon data collected in the Wood River system, Alaska (Carlson et al., 2007). These data are specific to the sites and years where they were collected; differences in water level, intensity of predation, and salmon abundance are all likely to change these values. For these reasons, future users of the method we demonstrated here should estimate stream life in their own systems, rather than relying on the model developed using the Carlson et al. (2007) data. This is particularly important because a sensitivity analysis showed our in stream salmon estimates were quite sensitive to changes in estimated stream life (Fig. 5); a two day increase in stream life increased the estimated maximum abundance by 14% on Meadow Creek and 22% on Southeast Creek.

Similarly, a good escapement estimate is only possible if users accurately model the relationship between time lapse and video counts (Hansen, Madow & Tepping, 1983). This is critical given the large differences in abundance estimates resulting from small differences in model structure or fit (Table 2 and Fig. 4). It is important to consider multiple model shapes; different stream morphologies or salmon species may produce different salmon run patterns. For example, steep streams are likely to produce models with different slopes for salmon swimming upstream and downstream. The segmented model structure can account for this pattern, and thus should always be included in the candidate model set. Also, a polynomial model might be appropriate for streams that experience high densities of spawners. In general, a polynomial model is needed if time-lapse detection of passing salmon changes with salmon run intensity. For example, as salmon reach high densities, they may not be able to move upstream very quickly because of crowding. This could result in relatively higher detection at high run intensities. In this case, a polynomial model would likely model the relationship better than a first order model. Regardless of the model shape, it is important that users use standard model diagnostics and good sense to fit the best model possible.

We have learned several important lessons from testing this method on different streams and different sites within streams. First, this counting system is most accurate and requires the least effort when located where flow is rapid but the water surface is smooth. The rapid flow prevents salmon from loitering above the contrast panels (which can introduce noise into the time-lapse counts), while the smooth water surface makes it easy to see passing salmon. Second, this system works best in shallow streams. Deep streams (>1 m) were problematic because salmon were more likely to swim at different depths, which caused their outlines to overlap and made counting more difficult. It was also more difficult to light deep streams at night. We found that our infrared lights did not light passing salmon adequately if streams were more than one meter deep. Using conventional flood lights (visible light) solves this problem; however, it negates the advantages of using IR lights, which is invisible to humans, fishes, and most wildlife. Third, it is important to orient the camera away from the sun (northward in the northern hemisphere), because otherwise the surface of the water reflects glare towards the camera.

Although this new method increases the breadth of sites that can be monitored, it has some limitations. As with other methods, the turbidity associated with high flow events can make seeing passing salmon difficult or impossible. Fortunately, these events tend to be brief in the small streams for which we designed this system. Also, it can be difficult to distinguish among species if a site has multiple species migrating at the same time. Finally, this system can only monitor streams up to 15 m wide. Beyond this width, counting accuracy is likely to decrease as the salmon in the images become more distant. One potential solution is to use two camera towers on opposite banks, each viewing one half of the stream.

Using this system, it can be difficult to accurately model the relationship between time-lapse counts and salmon passage if escapement is less than two or three thousand salmon. This is because at low escapement, hourly time-lapse counts tend to vary little, regardless of the relative intensity of the run. This makes it difficult to effectively model the relationship between time-lapse photo counts and video counts. One solution to this problem is to increase observer effort by either increasing the length of the sampling unit (e.g., from one to two hours) or by increasing the sampling frequency (e.g., 3-photo burst every 30 s). This would increase the contrast between weak and strong runs, but also increase the time required to review photos and/or video. Another solution is to use a model from a stream with similar features (width, depth, velocity, etc.), although we know from the data presented here that models can differ greatly among streams (Table 2). For example, if we had used the Southeast Creek model to estimate Meadow Creek escapement, we would have overestimated by 89% compared to the Meadow Creek top model.

In general, salmon researchers should strive to minimize their impact on natural salmon behavior. In small streams such as those monitored here, spawning salmon tend to move up and downstream frequently (Fig. 4, top), a behavior that may be a strategy for avoiding predators (Bentley et al., 2014). Salmon monitoring methods such as weirs have the potential to limit these movements. This could allow predators such as bears to catch salmon more easily, which could decrease salmon spawning success rates and alter trophic interactions with salmon consumers. A key strength of the method presented here is that it allows salmon to move freely and allows natural interactions with salmon consumers.

As with many resources used by wildlife, salmon availability is very patchy in space and time (Armstrong & Schindler, 2011). This presents a challenge for researchers and managers interested in using sampling to estimate their abundance; the more patchy or pulsed the salmon run, the less accurate a random sampling method will be without large amounts of effort. Here, we overcame this challenge by using a model-based design instead of a random sampling-based design. This allowed us to relax the demands on our camera system; rather than requiring complete video coverage, we merely needed hours of video that represented the full range of salmon run intensities. Given the ubiquity of patchy (in space) or pulsed (in time) resource availability, we suspect that this approach to double sampling could be employed in a variety of natural resources applications.

The salmon counting method that we present here expands the range of salmon spawning habitats that can be realistically monitored. Due to their low cost and relative portability, these systems would be ideal for monitoring salmon populations of conservation concern. For example, they could produce baseline and ongoing data on the abundance of salmon spawning downstream of mines or other resource development projects. Compared to existing methods, our solution is less expensive, less time consuming, and less detrimental to salmon and the wildlife that use them. The data produced can help improve our understanding of how population dynamics at small scales creates stability at the watershed scale.

Supplemental Information

Supplemental Information 1 R code used for all aspects of the analysis

All code used to conduct analysis and create figures. For simplicity, 5 separate r script files were combined. For best results paste each one into its own file before running.

Click here for additional data file.

Supplemental Information 2 Raw photo and video counts of sub-sampled hours

Hourly aggregated time-lapse salmon counts with corresponding video counts.

Click here for additional data file.

Supplemental Information 3 Raw time lapse salmon counts

Hourly aggregated time lapse salmon counts.

Click here for additional data file.

This work evolved from the early video enumeration efforts of Mat Sorum. His experimentation with remote video systems greatly accelerated the development of the methods presented here. We appreciate the staff at the Kodiak National Wildlife Refuge and Flathead Lake Biological Station for their committed support and assistance for this project. We thank Caroline Cheung, and volunteer field technicians Barbara Svoboda, Alex May, Bill Dunker, Tim Melham, Tyler Tran, Marie Jamison, Louisa Pless, Francesca Cannizzo, Isaac Kelsey, Mark Melham, Jane Murawski, Prescott Weldon, Shelby Flemming, Andy Orlando, and Kristina Hsu for their hard work in the field and lab. We thank pilots Kurt Rees and Kevin VanHatten for their skilled flying.

Additional Information And Declarations

Competing Interests

Author Contributions

Field Study Permissions

Data Availability

Jack Stanford is an Academic Editor for PeerJ.

William W. Deacy conceived and designed the experiments, performed the experiments, analyzed the data, wrote the paper, prepared figures and/or tables.

William B. Leacock conceived and designed the experiments, performed the experiments, contributed reagents/materials/analysis tools, reviewed drafts of the paper.

Lisa A. Eby analyzed the data, reviewed drafts of the paper.

Jack A. Stanford contributed reagents/materials/analysis tools, reviewed drafts of the paper.

The following information was supplied relating to field study approvals (i.e., approving body and any reference numbers):

The Alaska Department of Fish and Game gave permission for installation of camera systems by issuing permit FH-14-II-0076.

The following information was supplied regarding data availability:

The raw data has been supplied as a Supplemental Dataset.

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
