# Peer review of "A time-lapse photography method for monitoring salmon (Oncorhynchus spp.) passage and abundance in streams"

_PeerJ, doi:10.7717/peerj.2120_

## Round 0.1 · original submission · Minor Revisions

Both reviewers agreed that this is a useful paper that requires only minor revisions. I have also uploaded a pdf with some suggestions. My main request would be that you make it clear what R packages you used and to cite them. I was unable to download the R code you deposited.

Reviewer 1 ·

Basic reporting

Report meets the basic reporting requirements. I would suggest more recent literature cite on different monitoring techniques. See American Fisheries Society

Experimental design

No comments

Validity of the findings

no comments

Additional comments

Approximately 120000 photos were reviewed each month which seems like a large amount of work especially during the peak passage time during the middle of July.

Annotated reviews are not available for download in order to protect the identity of reviewers who chose to remain anonymous.

Reviewer 2 ·

Basic reporting

Overall the flow, organization and structure of the paper adhere to a standard research paper outline.
I do have some phasing, tense and rewording suggestions below:

Abstract

Over all the abstract is very well written and does a good job of conveying the scope of this study. Suggestions below.
Line 1: I would not start both the introduction and abstract with the same sentence
Line 13: I would also not include the cost and effectiveness of the camera technology in the abstract, include only an over view of the entire research, if you must put the cost in dollars I would add it to the discussion section and even then I would recommend reporting the cost in terms of the per portion of money saved for example "this system costs only 1/3 or less than these other systems."

Introduction

In general the Introduction does a good job of conveying the importance of the research and the benefits of its application to Salmon research, I do have some suggestions below.
Line 24: Rephrase "decide when to open and close fisheries" to "improve management timing of fisheries"
Line 27: I would cite a reference at the end of this sentence if you have one.
Line 31: Tense chance "at different times and in different locations tends to have more stable interannual abundance than," I would chance the tense on "tends" to "tend to have a more stable interannual..."
Line 34: Re phase "so it is important that we have the tools to investigate" to read "hence the importance of having the tools..."
Line 44: I would more evenly distribute the four citations at the end of this sentence throughout the paragraph, each one at the end of the sentence to of which it most pertains, similar to the sentence on lines 45 and 46.
Line 45: I wouldn't start this sentence with Also, just start with "Consumers are..."
Line 49: I would rephrase "because they are expensive, time consuming, and alter salmon behavior." to read "given high financial costs, investment of time and the potential to alter salmon behavior."
Line 68: Tense Change, "movements is common" to "are common."
Line 87: Again I recommend not including the cost of the technology in the paper unless in the discussion. It would be best in the supplemental materials.

Methods

Line 104: Tense change, We can determine to "we determined."

Results

Line 281: Change "an 5 day" to a 5 day and 2 day increase.

Discussion

Lines 301-304: Are the conditions, predators, river gradients and over all conditions etc similar to those of the wood river and the results shown by Carlson 2007?

Lines 380-383: The final paragraph does a good job summarizing the importance of the study but I would relocate the last two sentences to just after the first sentence of the last paragraph.


Suggestions for Tables and Figures:

Table one is fairly large and admittedly needs re adjustment, perhaps removal of the "manned" column would help with this. And removal would not deter from the overall message as it is somewhat self explanatory.
I would also recommend highlighting the row titles at the top of table two, similar to the column titles.
For consistency I suggest capitalizing the words "video counts" and "photo counts" in the second figure.
Lastly it may be worth creating all the figures with the same program, but then again this may not be possible.

Experimental design

All in all the experimental design seems well founded and technically sound. The fact that is has previous statistical founding adds more credence to the report. Methods are described clearly and the research question is clearly defined.

Below I have some concerns and suggestions.

Line 131: Does the majority of salmon movement and activity really take place between the hours of 12pm to 8pm? Do you have a reference for this?
Line 151: Why did you sample for a "non random" sub sample of hours, why not use a random sample of hours to attain less biased estimates?
Line 219: Were stream conditions measured in 2014 similar to the year this data was detected?
Line 238: Having an estimate of the number of consumers would not be bad given the effects a great number could have on the run, if it is available.

Validity of the findings

The findings presented in this research appear valid and statistically sound and controlled. Data is provided and conclusions are well stated. Speculation is identified wherever present.

I would still stress the importance of not including the cost of the camera tech in the paper as this research should hopefully not be used as a tech selling platform.

Additional comments

All in all I was very impressed by this research paper and wish the authors luck with its publication.

---

## Round 0.2 · accepted · Accept

Great to see this one going through to publication. I have suggested some really minor edits that hopefully PeerJ staff can just implement:

Line 51, space after full stop needed.
Line 202, reference for segmented should be (Muggeo, 2008)
Line 218, reference for boot should be (Canty and Ripley, 2016)

Check reference Kohavi R (1995). Needs a newline I think.

Add to reference list (I generated these using citation("package").

Muggeo, VMR (2008). segmented: an R Package to Fit Regression Models with Broken-Line Relationships. R News, 8/1, 20-25. URL http://cran.r-project.org/doc/Rnews/
Canty, A and Ripley, B (2016). boot: Bootstrap R (S-Plus)
Functions. R package version 1.3-18.